# RETRACTED: The Brain Penetrating and Dual TORC1/TORC2 Inhibitor, RES529, Elicits Anti-Glioma Activity and Enhances the Therapeutic Effects of Anti-Angiogenetic Compounds in Preclinical Murine Models

**DOI:** 10.3390/cancers11101604

**Published:** 2019-10-21

**Authors:** Giovanni Luca Gravina, Andrea Mancini, Alessandro Colapietro, Simona Delle Monache, Roberta Sferra, Simona Pompili, Flora Vitale, Stefano Martellucci, Francesco Marampon, Vincenzo Mattei, Leda Biordi, David Sherris, Claudio Festuccia

**Affiliations:** 1Department of Biotechnological and Applied Clinical Sciences, Division of Radiation Oncology, University of L’Aquila, 67100 L’Aquila, Italy; giovanniluca.gravina@univaq.it; 2Department of Biotechnological and Applied Clinical Sciences, Laboratory of Radiobiology, University of L’Aquila, 67100 L’Aquila, Italy; mancio_1982@hotmail.com (A.M.); alecolapietro@gmail.com (A.C.); 3Department of Biotechnological and Applied Clinical Sciences, Laboratory of Cellular Biology, University of L’Aquila, 67100 L’Aquila, Italy; simona.dellemonache@univaq.it (S.D.M.); s.martellucci@sabinauniversitas.it (S.M.); 4Department of Biotechnological and Applied Clinical Sciences, Laboratory of Human Anatomy, University of L’Aquila, 67100 L’Aquila, Italy; roberta.sferra@univaq.it (R.S.); pompili.simona@virgilio.it (S.P.); 5Department of Biotechnological and Applied Clinical Sciences, Laboratory of Neurophysiology, University of L’Aquila, 67100 L’Aquila, Italy; floravitale86@Hotmail.it; 6Laboratory of Experimental Medicine and Environmental Pathology, University Hub “Sabina Universitas”, 02100 Rieti, Italy; Vincenzo.mattei@uniroma1.it; 7Department of Radiological, Oncological and Pathological Sciences, Sapienza University of Rome, 00118 Rome, Italy; francesco.marampon@uniroma1.it; 8Department of Biotechnological and Applied Clinical Sciences, Laboratory of Experimental Oncology, University of L’Aquila, 67100 L’Aquila, Italy; leda.biordi@univaq.it; 9Sherris Pharma Partners, Jamaica Plain, MA 02130, USA; davidsherris@gmail.com

**Keywords:** glioblastoma, RES529, TORC1/TORC2 inhibitor, bevacizumab

## Abstract

**Background.** Glioblastoma multiforme (GBM) is a devastating disease showing a very poor prognosis. New therapeutic approaches are needed to improve survival and quality of life. GBM is a highly vascularized tumor and as such, chemotherapy and anti-angiogenic drugs have been combined for treatment. However, as treatment-induced resistance often develops, our goal was to identify and treat pathways involved in resistance to treatment to optimize the treatment strategies. Anti-angiogenetic compounds tested in preclinical and clinical settings demonstrated recurrence associated to secondary activation of the phosphatidylinositol 3-kinase (PI3K)/AKT/mTOR pathway. **Aims.** Here, we determined the sensitizing effects of the small molecule and oral available dual TORC1/TORC2 dissociative inhibitor, RES529, alone or in combination with the anti-VEGF blocking antibody, bevacizumab, or the tyrosine kinase inhibitor, sunitinib, in human GBM models. **Results.** We observed that RES529 effectively inhibited dose-dependently the growth of GBM cells in vitro counteracting the insurgence of recurrence after bevacizumab or sunitinib administration in vivo. Combination strategies were associated with reduced tumor progression as indicated by the analysis of Time to Tumor Progression (TTP) and disease-free survival (DSF) as well as increased overall survival (OS) of tumor bearing mice. RES529 was able to reduce the in vitro migration of tumor cells and tubule formation from both brain-derived endothelial cells (angiogenesis) and tumor cells (vasculogenic mimicry). **Conclusions.** In summary, RES529, the first dual TORC1/TORC2 dissociative inhibitor, lacking affinity for ABCB1/ABCG2 and having good brain penetration, was active in GBM preclinical/murine models giving credence to its use in clinical trial for patients with GBM treated in association with anti-angiogenetic compounds.

## 1. Introduction

Glioblastoma (GBM—grade IV glioma) is the most aggressive primary brain tumor. Despite aggressive surgical and chemo/radiotherapies, this neoplasia shows high rate of therapeutic failures and poor prognosis [1]. GBM is characterized by aggressive progression and mean overall survival (OS) of 14.6 months [2]. Failure of standard chemo/radiotherapy at least in part may be due to microenvironment protection, de novo and/or acquired tumor resistance, limitations in drug delivery, increased angiogenesis and/or vasculogenic mimicry (VM), and presence of glioma stem cells (GSCs) [3]. For this reason, alternative approaches that can target the mechanisms of recurrence, would be of value for therapeutic treatment of GBM. GBM is well known for its elevated vascularity [4] and agents targeting neoangiogenesis were considered as alternative therapeutic strategies for this disease [4]. Increased vascular endothelial growth factor (VEGF) expression was detected in GBMs, and its elevation was associated with poor prognosis [5,6]. Bevacizumab (Avastin) is a recombinant humanized monoclonal antibody to VEGF that inhibits its binding to its receptor [6] and was the first angiogenesis inhibitor that has demonstrated to improve overall response rates, time to progression, and survival in cancers [7]. Bevacizumab was approved by the FDA in 2009 [8] to treat GBM recurrence on the basis of encouraging data [9,10,11]. Bevacizumab reduces tumor edema, angiogenesis, and disease burden but these effects are followed by resistance as a result of adaptive tumor responses [12,13] such as modifications of the perivascular niche [14,15,16,17], activation of different growth factors [15,16,17,18], increased hypoxia [19,20], and recruitment of circulating monocytes and macrophages [19]. GBM resistant tumors appear with local invasion and more diffuse borders [20] suggesting that resistance may be characterized by a more invasive state [21,22]. Despite unsatisfactory clinical data, some patients with recurrent GBM (rGBM) treated with bevacizumab or tyrosine kinase inhibitors (e.g., sunitinib [23,24]), showed an improved six-month progression free survival rate (PFS), and radio-graphic responses [25,26] which are not to be underestimated. The PI3K/Akt/mTOR axis [27,28] was identified as the major signaling pathway frequently altered in GBM [28]. Amplification and/or mutation of the EGFR gene, mutation and/or deletion [29] of the PTEN gene and mutations of the PIK3CA gene are in up to 80% of GBM patients [28,30]. Moreover, pathways associated with a variety of growth factors have been shown to be associated to GBM progression [31]. These pathways participate to the PI3K/Akt/mTOR activity regulating in turn tumor cell growth/viability, drug resistance, and angiogenesis [32,33] through p70S6 kinase (p70S6K), ribosomal S6 protein (RPS6), and eukaryotic initiation factor 4E-binding protein 1 (4EBP1) [34,35]. For this reason, it was thought that Akt driven tumors may be sensitive to the TORC1 dissociative inhibitor rapamycin (RAPA) or its analogs [36,37]. However, limited efficacy in clinical trials [38], led to their non-use in GBM as evidenced in ClinicalTrials.gov. As opposed to TORC1 inhibition, TORC2 may have a major role in promoting GBM growth or mediating chemotherapy resistance [39,40]. From this data, dual TORC1/TORC2 inhibitors could then represent a useful therapeutic approach. One such of these inhibitors is the small molecule 8-(1-Hydroxy-ethyl)-2-methoxy-3-(4-methoxy-benzyloxy)-benzo[c]-chromen-6-one named RES529 (previously Palomid 529 or P529) RES529 markedly reduces the signaling mediated by phosphorylation of Akt (S473) [41,42,43,44] and in preclinical in vivo murine models was able to inhibit angiogenesis, vascular permeability, and tumor growth [41,44]. RES529 was also shown to cross the blood-brain barrier (BBB). This requires a tightly regulated trans-endothelial trafficking mediated by transporters including the ATP-binding cassette (ABC) drug efflux transporters that are expressed at the luminal side of the brain micro-vascular endothelial cells [45,46]. RES529, indeed, is not a substrate for ABCB1/ABCG2 [44]. Here, we studied the effectiveness of RES529 alone or in combination with bevacizumab or sunitinib in three human GBM cell lines grown as subcutaneous in vivo models, luciferase transfected U87MG, and a patient derived glioma stem-like cell line GSCs-5 injected orthotopically in the brain. RES529 inhibited GBM cell proliferation/viability, induced apoptosis, and inhibited tubule formation and vasculogenic mimicry. When combined with standard anti-angiogenetic treatments, RES529 strongly inhibited tumor growth and angiogenesis. RES529 was further tested in a mice intracranial model of GBM. Animals treated intracranially with RES529 had significantly extended survival compared to control animals. RES529 effectively inhibits growth of the orthotopic U87MG glioblastoma and the CSCs-5 patient derived glioma stem-like model, synergizing with bevacizumab. Both models showed prolonged disease-free survival (DSF) and overall survival (OS) of the tumor bearing mice.

## 2. Material and Methods

### 2.1. Reagents and Drugs Preparation

Materials for tissue cultures were purchased from Euroclone (Milan, Italy). Antibodies against p-Akt Ser473 [sc-135651], p-Akt Thr308 [sc135650], p-Ser2448 mTOR [sc-293133], p-Ser 37/46-4E-BP1 [sc-293124], and pSer235/23-S6 [sc-293144] were purchased from Santa Cruz (Santa Cruz, CA, USA). MIB1 (Ki67) antibody was purchased from Dako (Carpenteria, CA, USA). Autophagy assay kit (ab139484) was purchased from Abcam (Cambridge, UK). ApopTag^®^ peroxidase in situ apoptosis detection kit was purchased from Merck Millipore (Milan, Italy). RES529 was provided by Diffusion Pharmaceuticals, Inc. Bevacizumab and sunitinib (Suni) were obtained from the pharmacy of the San Salvatore hospital in L’Aquila. RES529, bevacizumab, and sunitinib were used as described [16,18,42]. Necrostatin-1 [Methylthiohydantoin-DL-tryptophan, CAS 4311-88-0, 51] and the ferrostatin-1 were purchased from Sigma Aldrich (St Louis, CA, USA).

### 2.2. Cell Lines and Cell Cultures

Twelve human glioma cell lines (U251MG, U373, U118, U138, A172, U87MG, SW1783, SNB19, LN229, T98G, SF268, and D54) were maintained in DMEM containing 10% (*v*/*v*) fetal bovine serum, 4 mM glutamine, 100 IU/mL penicillin, and 100 μg/mL streptomycin. To minimize the risk of working with misidentified and/or contaminated cell lines, cells were stocked at very low passages and used <20 subcultures. However, periodically, a DNA profiling was carried out [16]. Luciferase transfected U87MG cells were kindly provided by JE Heikkila (Abo Akademi University, Turku, Finland). Six GBM patient-derived stem cell lines (BT12M, BT25M, BT48EF, BT50EF, BT53M), were kindly provided by J.G. Cairncross and S. Weiss (University of Calgary, Canada) [47], and GSCs-5 and CSCs-7 [48] from M. Izquierdo (Universidad Autónoma de Madrid, Spain). Luciferase was inserted in the genome of GSCs-5 cells using the pGL4.13 vector (Promega, Milan, Italy) and the jetPEI DNA transfection method (Polyplus, Illkirch, France). For negative controls, hBMEVC (human brain microvascular endothelial cells) were used, kindly provided by Philip M. Cummins (School of Biotechnology, Dublin City University, Ireland) and cultured in endothelial medium. These cells were routinely grown in complete endothelial cell medium (EGM™-Plus, Lonza Biologics, Slough, UK) containing heparin (0.75 U/mL), hydrocortisone (1 µg/mL), recombinant human epidermal growth factor (5 ng/mL), EndoGRO-LS Supplement (0.2%), and antibiotics. For positive controls, hBMVEC was used with administration of 10 ng/mL VEGF triggering robust Akt activity.

### 2.3. Cell Viability, Cell Cycle, and Apoptosis Analyses

RES529 effectiveness was evaluated by using the cell counting kit-8 (CCK-8; Dojindo Molecular Technologies Inc., Tokyo, Japan) calculating IC50 values by Grafit software (Erithacus, Wilmington House, UK). Neurosphere growth were verified by direct count of sphere originated by single cell suspensions at different times as described [49]. Data on cell cycle and apoptosis were collected by using Propidium chloride and Alexa Fluor^®^ 488 Annexin V/Dead Cell Apoptosis Kit (Life Technologies Europe BV, Monza, Italy) as described [16]. Apoptosis was also evaluated measuring the activity of caspase-3, caspase-8, and caspase-9 by using specific colorimetric substrates such as N-zDEVD-pNA, Ac-Ile-Glu-Thr-Asp-pNA, and Ac-Leu-Glu-His-Asp-pNA, respectively.

### 2.4. Cell Lysates and Enzymatic Analysis

Following treatments, cells grown in 90 mm diameter Petri dishes, were washed with cold PBS and immediately lysed with 1 mL lysis buffer containing a proteinase and phosphatase inhibitor cocktail. Total lysates were used for enzymatic evaluation of p-Ser473 Akt, p-Thr408 Akt, p-Ser2448 mTOR, Thr46/47-4E-BP1, and pSer235/236-S6 by ELISA. For the quantification of fluorimetric units (RFU), we followed instruction of manufacturers to subtract RFU values for each cell line values of reactive blacks and negative controls generated without flurochrome or cells. In addition, we normalize fluorimetric units thus obtained with GAPDH used as a housekeeping gene. We analyzed the fluorescence by using the TECAN GENios Multifunction reader (Tecan Italia, Cernusco sul Naviglio MI, Italy).

### 2.5. Glioblastoma Xenograft Model

Six-week-old female CD1-nu/nu mice were purchased from Charles River (Milan, Italy) and maintained under the guidelines established by the European Community (EC) guidelines 2010/63/UE and DL 26/2014 references. All mice received 1 × 10^6^ GBM cells injected in the flank. Tumor growth was assessed bi-weekly with a Vernier caliper and volumes calculated through the use of the ovoid formula, 4/3πR1xR2xR3 [16,18] where R1 is the radius relative to the width, R2 is the radius relative to the length, and R3 is the one relative to the thickness. In a first experiment, and about 10 days after tumor injection, 20 mice with tumor volumes of 0.8–1.3 cm^3^ were randomly divided into four groups (five mice per group with two tumors each, as described in Table 1): (1) Control (vehicle); (2) RES529 (25 mg/kg/5 day/week, by oral gavage—“per os” or PO); (3) RES529 (50 mg/kg/5 day/week, PO); (4) RES529 (100 mg/kg/5 day/week, PO). U87MG, A172, and T98G were used in xenograft models. In a second set of experiences, 30 mice were randomly divided into six groups (five mice per group with two tumors each): (1) Control (vehicle); (2) bevacizumab 4 mg/kg intraperitoneally (i.p.) every 14 days; (3) RES529 (50 mg/kg/5 day/week, per os (PO)); (4) BEV plus RES529; (5) sunitinib, 40 mg/kg PO quaque die (qd/every day); (6) SUNI plus RES. U87MG and A172 were used in xenografts models. At the end of the experiments, animals were sacrificed by carbon dioxide inhalation and tumors were subsequently removed surgically. Tumors were collected for protein and immunohistochemical analyses as described [49]. In order to reduce the probability of bias due to differences in tumor engraftment, tumor progression was analyzed by the parameter “Time to Progression (TTP)”, defined as the time (days) necessary to double the tumor volume for each tumor. The percentage of animals in progression was plotted in time by using Kaplan-Meyer distribution as previously described [16,18,49,50].

### 2.6. Immunohistochemical Analyses

Indirect immunoperoxidase staining was performed on paraffin-embedded tissue sections (4 μm). A consensus judgment as indicated in our previous report [33] was adopted as to the proper immunohistochemical score of the tumors based on the strength of positivity: Negative (score 0), weak (score 1), moderated (score 2), or strong staining (score 3). In each category, the percentage of positive cells was assessed by scoring at least 1000 cells in the area with the highest density of antigen positive cells. Cytoplasmic/membrane staining intensity was graded as follows: 0 = negative; 1 = less of 10% of positive cells; 2 = positive cells in a range of 10–50%, and 3 = more than 50% positive cells. Overall expression was defined by the staining index (SI) and ranged between 0 and 9 with an SI ≤ 4 indicating a low expression. Proliferation index (labeling index) was determined though the evaluation of percentage Ki67 positive cells analyzing 500 cells at 100× magnification. TACS Blue Label kit (code 4811-30-K, R&D Systems, Inc., Minneapolis, MN, USA) was used for apoptosis in situ determination and the tunnel positive cells percentage determined on five random fields evaluated at 400×. In order to count the number of CD34+ microvessels five arbitrarily selected fields, analyzed for each group at 100×/microscopic field (tumor microvessels), were used. For CD34 staining, we used purified anti-mouse CD34 Antibody (MEC14.7) and anti-human CD34 antibody (clone 581) purchased from Biolegend (London, UK).

### 2.7. Orthotopic Intra-Cranial Model

Female CD1 nu/nu mice were inoculated intra-cerebrally as previously described [16,18,51] by using luciferase transfected U87MG and GSCs-5. Five days after injection, when no luciferase activity was intracranially detectable, animals were randomized on 10 mice/group. In a first experiment type, we divided animals into four groups (Table 1): (1) Control (vehicle); (2) RES529 (25 mg/kg/5 day/week, PO); (3) RES529 (50 mg/kg/5 day/week, PO); (4) RES529 (100 mg/kg/5 day/week, PO). Next, the dose of 100 mg/kg of RES529 was tested in combination with bevacizumab or sunitinib: Sixty mice were randomly divided into six groups as described above (ten mice per group, Table 1): (1) Control (vehicle); (2) bevacizumab (BEV) 4 mg/kg PO every 14 days; (3) RES529 (100 mg/kg/5 day/week, PO); (4) BEV plus RES529; (5) sunitinib (SUNI, 40 mg/kg PO qd); (6) SUNI plus RES529 in vivo bioluminescence images were obtained using the UVITEC Cambridge Mini HD6 (UVItec Limited, Cambridge, UK). Animals were anesthetized and luciferin (150 mg/kg) was injected intra-peritoneally 15 min prior to imaging. Images were collected and bioluminescence intensity (BLI) was measured. To simulate pathological condition at the time for which surgery occurs (low number of tumor cells in wound bed causing regrowth recurrence), we inoculated a small number of cells (3 × 10^3^) and started drug administration when no bioluminescence was detected. Mice were euthanized when they displayed neurological signs or weight loss of 20% or greater to record data on overall survival (OS).

### 2.8. Matrigel Plug Assay

Liquid Matrigel (10–16 mg of protein/mL) was mixed at 4 °C with 10 ng/mL VEGF dissolved in PBS at a final concentration equal to 10.0 µg/mL and injected subcutaneously (0.5 mL/mouse) into the flank of 6–8 week old balbc mice (Charles River, Calco, Italy). Matrigel with PBS alone was used as negative control. Treatment with RES529 was compared to 4 mg/iv bevacizumab administered at time 0. One week after injection, mice were sacrificed and plugs were harvested, weighted, and divided in two parts, one half underwent for biochemical analyses (i.e., hemoglobin assay) whereas the other half was processed for IHC and biochemical analysis. Each Matrigel plug was collected, fixed in 4% formalin, paraffin embedded, cut, stained with HE, and photographed.

### 2.9. Statistics

Numeric data are expressed as mean ± SD or median and 95% CI. ANOVA was followed by Tukey’s test. TTP, DFS, and OS were analyzed by Kaplan-Meyer curves and Gehan’s generalized Wilcoxon test. *p* values < 0.05 were considered statistically significant. SPSS^®^ (statistical analysis software package) version 10.0 and StatDirect (version. 2.3.3., StatDirect Ltd., Birkenhead, UK) were used for statistical analysis and graphic presentation.

## 3. Results

In Figure 1A we demonstrate that p-Ser473 Akt, p-Thr408 Akt, p-Ser2448 mTOR, and Thr46/47-4E-BP1 levels were highly expressed in GBM cell models, whereas in Figure 1B RES529 inhibited -Ser473 Akt (upstream enzyme) and pSer235/236-S6 (downstream enzyme) with similar IC50 values in U87MG (as a model for differentiated GBM cells) and CSCs-5 (as a model of cells with stem-like phenotype). In order to have reference cell lines as controls, we added a “negative control” represented by human brain derived normal endothelial cells (hBMVEC) which at the basal level, in absence of angiogenic stimuli such as VEGF, are negative for Akt activation and a second “positive control” represented by the same endothelial cells administered with 10 ng/mL VEGF. In this case, hBMVEC forms in Matrigel tubule structures similar to vessels after triggering a robust Akt activity. GBM cell lines and VEGF treated hBMVEC showed significantly higher Akt/mTOR activity (Figure 1A) versus the negative control (hBMVEC without VEGF) and cutoff of ELISA determinations.

### 3.1. Antiproliferative Effects of RES529

The growth inhibitory effects of RES529 was evaluated in twelve GBM cell lines and seven patient-derived GSCs. We demonstrated that RES529 displayed a dose-dependent response in suppressing glioma cell survival (Figure 1C) with mean IC50 value of 9.9 ± 6.2 μM (range: 2.0 to 30.5 μM). No differences were observed between established GBM (9.6 ± 7.7 μM) and stem-like cells (6.3 ± 3.5 μM) and although mTOR activation is important for cell cycle progression [23,24], the efficacy of RES529 seemed not to be related to PI3K/Akt/mTOR basal phosphorylation levels. In order to verify whether normal endothelial cells were sensitive to RES529, we used hBMVEC untreated or stimulated with VEGF in order to trigger a robust Akt activity.

As seen in Figure 1C, untreated hBMVEC was not sensitive to RES529, if not at the highest doses with IC50 > 50 µM, whereas VEGF-stimulated hBMVEC resulted as sensitive to RES529 with IC50 < 20 µM. A172, U87MG, U251, and T98G cells were selected for further analyses. These cells are also widely known to undergo proliferation, autophagy, drug sensitivity, and apoptosis and were previously used in order to evaluate their sensitivity to PI3K, Akt, and mTOR inhibitors. Therefore, it is possible to make comparisons with these compounds when used with new compounds as in RES529. In addition, these cells have been shown to be sensitive to RES529. The propidium iodide/annexin V assays and Ki67 immunostaining were used to analyze apoptosis and cell cycle. Integrating five replicates of FACS assays, RES529 treatment (at IC20 values) in glioma cells increased the subG1 cell percentage and Go/G1 phase cells (Figure 1D). In Appendix A we show the FACS analyses performed in A172 cell model. In addition, APOSTRAND™ ELISA apoptosis detection kit (Appendix A) was used to have data on apoptosis. In Appendix A we show the cell cycle distribution observed in A172 cells with an initial increment in Go/G1 cell cycle phase at IC20. At IC50 RES-529 dose subG1 cells increase significantly reducing the percentage of cells in G0/G1 and increasing the percentage of cells in G2/M. So, a dual effect in G1 and G2 phases are observed. Sub G1 increment was associated to apoptosis whether we consider both the increment of densitometric values from APOSTRAND™ ELISA apoptosis detection kit and caspase-3 activity. RES529 was able, indeed, to induce caspase-3 activity at their IC20 and IC50 for each cell line (Appendix A). Significantly, indeed, z-VAD-fmk, the pan-caspase inhibitor, largely inhibited RES529 induced caspase-3 activity (Appendix A). These results indicate that RES529 induces glioma cell apoptotic death.

### 3.2. RES529 In Vivo Monotherapy

RES529 was evaluated for anti-tumor effects in vivo at three dose levels (25, 50, and 100 mg/kg/days). In Figure 2, the effects of RES529 is shown as monotherapy using the U87MG subcutaneous xenograft model. Experimental conditions and treatments are shown in Figure 2A (see also Table 1).

We show that RES529 reduced the weight of tumors 19.3% (1091 mg ± 399 in the control versus 880 mg ± 302), 37.3% (684 mg ± 231), and 62.2% (412 mg ± 110) after administration of 25, 50, and 100 mg/kg/day, respectively (Figure 2B and Appendix A). Similarly, (Figure 2C) TTP values were not significant for the treatment at 25 mg/kg/day (with similar TTP values versus control) whereas the differences became significant for the 50 and 100 mg/kg dose levels (Appendix A). Kaplan-Meyer curves demonstrate that 50 and 100 mg/kg/day statistically reduced tumor progression of U87MG tumors with hazard ratios of 2.73 and 3.93, respectively (Figure 2D and Appendix A). This experiment was repeated using A172 (Appendix A) and T98G (Appendix A) xenografts confirming the data from U87MG xenografts. In particular, RES529 reduced A172 tumor weight of 27.8%, 52.1%, and 71.1% after administration of 25, 50, or 100 mg/kg/day, respectively. Differences for TTP were statistically significant for all dose levels. Kaplan-Meyer curves demonstrated that all dose levels reduced the tumor progression of U87MG tumors with hazard ratios ranging between 2.15 and 5.26. RES529 also reduced T98G tumor weights corresponding to 15.4%, 32.2%, and 57.7% after administration of 25, 50, or 100 mg/kg/day, respectively. The difference in tumor weight and TTP were statistically significant only for the 50 and 100 mg/kg/day dose levels. Kaplan-Meyer curves demonstrate, instead, that all dose levels reduced tumor progression of T98G tumors with hazard ratios ranging between 2.48 and 4.54.

### 3.3. RES529 Increased Disease-Free and Overall Survival in Orthotopic Intra-Brain Tumours

As described in section MM, a small number of cells (3 × 10^3^), were inoculated into the brain in order to mimic a clinical situation in which a small residual mass remain after surgery. In Figure 3A we show a schematic protocol for the dose-dependent experiment with doses of 25, 50, and 100 mg/kg/day (see also Table 1). In Figure 3B and in Appendix A we demonstrate that the time necessary to detect by bioluminescence in the intra-brain tumor, increased after RES529 administration and was incremented in a dose-dependent manner. In particular, control mice developed a bioluminescent lesion after 15–37 days with a mean of 24.0 ± 8.43 days. In RES529 (25 mg/kg/day) treated animals, mice developed a bioluminescent lesion after 20–30 days with a mean of 24.5 ± 3.69 (*p* = 0.8655 (NS) versus control). DFS ranged between 20–45 days, with a mean of a 30.0 ± 7.45 (*p* = 0.0691 (NS) versus control, after 50 mg/kg/day RES529. Statistically significant differences were observed when 100 mg/kg/day of RES529 were administered to mice. There, mice developed lesions after 25–60 days with a mean of 40.0 ± 14.53 (*p* = 0.0075 versus CTRL; *p* = 0.0043 versus 25 mg/kg/day and *p* = 0.0687, NS, versus 50 mg/kg/day. Evaluation through Kaplan-Meyer curves (Figure 3C) and HR values (Appendix A) showed that administration of 100 mg/kg/day had greater effects indicating optimal dosage in the orthotopic intra-brain models. Analysis of Kaplan-Meyer curves further demonstrated that 100 mg/kg/day resulted in significant hazard ratio versus control (2.59) and 25 mg/kg (2.49) but not versus 50 mg/kg/day (1.99). In Figure 3D and Appendix A, we found that the OS values in control mice ranged between 30 and 73 days with a mean of 48.40 ± 17.64 days. The dose of 25 mg/kg/day increased OS to 37 and 65 days with a mean of 50.40 ± 9.33. Administration of 50 mg/kg/day increased OS up to 44 and 78 days with a mean of 61.10 ± 13.30. These differences were not statistically significant for both versus CTRL and between them. OS shown in animals treated with 100 mg/kg/day of RES529 ranged between 40 and 120 days with a mean of 86.0 ± 28.65 and was statistically significant versus CTRL and the other doses. Through Kaplan-Meyer curve (Figure 3E) analysis of HRs were calculated for each experimental group (Appendix A). Administration of 100 mg/kg/day show significant protective effects versus the other two dose levels indicating this was the optimal dosage in orthotopic intra-brain models. Analysis of Kaplan-Meyer curves demonstrated that 100 mg/kg/day reduced the percentage of animal progression of U87MG tumors with a hazard ratio range between 3.07 versus control, 2.44 (versus 25 mg), and 2.49 (versus 50 mg/kg/day).

### 3.4. RES529 Blocks Angiogenesis and Vasculomimicry

Since glioblastomas are highly vascularized tumors, Akt/mTOR pathways are essential for VEGF-induced angiogenesis. RES529 was shown to reduce tumor angiogenesis and vascular permeability [41]. Additionally, vasculogenic mimicry may play a role in escaping antiangiogenetic therapies. For this reason, RES529 was evaluated for its ability to block such processes. RES529 inhibited VEGF-dependent cell proliferation and tubule formation of brain derived endothelial cells (hBMVEC, Figure 4A) with similar efficacy of bevacizumab (positive control). The reduction of cell proliferation and tubule formation showed similar IC50 of 5.02 and 8.23 µM, respectively (Figure 4B). RES529 was analyzed in vivo at 50 mg/kg/day for 10 days to determine if it could inhibit the influx of vessels in a Matrigel plug assay. In Figure 4C, we show a hematoxylin/eosin stain of different Matrigel plugs demonstrating the colonization of each plug with murine stromal cells (vehicle, negative control) as well as the predisposition of these cells to form large vascular structures. The presence of 10 ng/mL endothelial cell growth factor (VEGF), dissolved in the Matrigel, resulting in maximal level of induction of vascular structures representing the positive control. Administration of RES529 in nude mice reduced both the size and number of vascular structures suggesting an anti-angiogenetic effect of RES529. The measure of this effect was quantified by the evaluation of hemoglobin content performed in frozen plugs (Figure 4D). Angiogenesis was reduced after administration of RES529. Vessel counts and measurements of hemoglobin, extracted from individual plugs, gave a measure of the anti-angiogenic synergistic effect (Figure 4D).

When tested in vitro, RES529 confirmed its action as an inhibitor of VM by significantly reducing the “vessel-like formation” promoted by several glioblastoma cell lines, including U87MG and GSCs5 cells whereas bevacizumab had a poor effect on this pathological behavior. In Appendix A, we show the vasculomimicry obtained in GSCs-5 cells consistent with its inhibitory potency of the inhibition of PI3K/Akt/mTOR pathways [52]. Since GBM is highly invasive in nature and frequently invades normal brain parenchyma, it is difficult to get complete resection. The migratory/invasive capacity was analyzed showing that RES529 was able to reduce invasion and migration of different GBM cell types. Reduction of VM by RES529 was associated by reduced migration and Matrigel invasion (Appendix A). This suggests that RES529 significantly affected both angiogenesis and VM processes and that bevacizumab resulted in a synergic inhibition of U87MG VM. This supports the use of dual administration as a therapeutic modality. It is reasonable, then, that a therapeutic regimen consisting of both RES529 and antiangiogenic compound could result in a further increase in therapeutic outcome in vivo. For this reason, we chose to evaluate RES529 in the subcutaneous model and the orthotopic intra-brain model.

### 3.5. RES529 Shows Additive Effects with Anti-Angiogenetic Therapy In Vivo: (Subcutaneous Xenograft Model)

To this aim, U87MG (Figure 5 and Appendix A) and A172 cells (Appendix A) glioblastoma cell lines were subcutaneously injected in female nude mice as described and animals were randomly distributed in six experimental groups, as described in Figure 5A and Table 1. 50 mg/kg/day RES529 was confirmed to exert a significant inhibitory effect on the growth of human tumor grafts compared to the control and resulted comparably to bevacizumab. Administration of RES529 significantly decreased tumor weight from 909 ± 179 mg to 457 ± 113 mg (49.7%, Appendix A) in A172 xenografts and from 756 ± 164 mg to 570 ± 123 mg (24.7%, Figure 5 and Appendix A) in U87MG ones. Bevacizumab reduced tumor weight of about 50% in A172 (445 ± 170 mg) and 37% in U87MG (480 ± 150 mg) xenografts. Combination of bevacizumab and RES529 overcomes the limits of anti-VEGF based treatment in GBM xenografts. In our experimental conditions, RES529 significantly increased the efficacy of bevacizumab, reducing tumor growth further. As an example, A172 (Appendix A) xenografts treated with RES529 plus bevacizumab reached a final volume of 281 ± 76 mg (about 70% versus CTRL) whereas U87MG xenograft passed from 756 ± 164 mg to 179 ± 23 mg (76% versus CTRL, Figure 5B and Appendix A).

Shrinkage of the tumor mass induced by oral administration of RES529 in combination with bevacizumab resulted in a significant increase of the Time to Progression (TTP) in A172 of about 116% (21.60 ± 3.50 days) versus CTRL and 69% and 50% versus RES529 and bevacizumab monotherapy, respectively. Similarly, in U87MG xenografts TTP passed from 12.20 ± 2.39 days to 22.60 ± 3.53 that is +85% versus CTRL and +52% versus bevacizumab and +47% versus RES529 monotherapy. Sunitinib reduced tumor weight of about 50% in A172 (462 ± 143 mg) and U87MG (378 ± 116 mg) xenografts. RES529 significantly increased the efficacy of sunitinib in both xenografts reaching a final volume of 230 ± 72 mg (about 75% versus CTRL in A172 xenografts whereas U87MG xenografts passed to 160 ± 72 mg (79% versus CTRL). Shrinkage of the tumor mass induced by oral administration of RES529 in combination with sunitinib resulted in a significant increase of the Time to Progression (TTP) in A172 of about 135% (23.55 ± 3.67 days) versus CTRL and 83% and 85% versus RES529 and sunitinib monotherapy, respectively. Similarly, in U87MG xenografts TTP passed from 12.20 ± 2.39 days to 22.60 ± 3.53 that is +85% versus (Figure 5C) CTRL and +40% versus both sunitinib and RES529 monotherapies. Kaplan-Meyer analyses (Figure 5D, Appendix A) indicated that combination of RES529 with bevacizumab and sunitinib were additive or weakly synergistic with CI ranging between 0.92 and 1.04 and hazard ratios were statistically significant when compared with single treatments.

### 3.6. Histological Analyses

Next, we analyzed the morphological changes after the association of RES529 (50 mg/kg os qd) and bevacizumab (4 mg/kg iv every 14 days) on mice xenografts mainly to assess modification in vessel network and vasculogenic mimicry (VM). It has been widely demonstrated that after an initial antitumor effect, which “normalize” the angiogenetic network, bevacizumab induced the triggering of protective pathways able to modulate the vasculogenesis which takes place to angiogenesis. Usually elevated angiogenesis and vasculogenesis may be observed in untreated tumors for the presence of CD34 positive cells which react with murine isotype (angiogenesis) or with the human isotype (human tumor vasculo-mimetic cells). In Figure 6, we show human CD34+ cells are expressed in tissues derived from untreated tumors (vehicle) as single dispersed cells commonly not associated with organized vessels, however, murine CD34+ cells form vascular structures in the tumor parenchyma. Periodic acid-schiff (PAS) staining on xenograft histological specimens showed that angiogenesis and vasculogenic mimicry are equally distributed in untreated U87MG xenografts. Furthermore, we show that the structure of the entire vessel was straightened by a dense PAS positive connective matrix. In the “vasculo-mimetic” region of the wall, a series of small vascular lacunae (micro-capillaries were also present). The number of these structures is considerably higher in these PAS positive enclosed area compared to those formed in the “angiogenic” portion of the tumor. Histological slides from bevacizumab-treated animals, showed a significant reduction of murine CD34 positive vessels. As previously demonstrated [16,18], bevacizumab induced vasculomimicry associated with murine endothelial cells death and proliferation of human CD34 positive cells with envelope preformed vessel structures.

Vasculogenic mimicry start from single medium-sized vessels. In Figure 6, we observe that endothelial cells delimitate nearly the majority of the vascular wall whereas in a very little portion of it, a loss of endothelial cell continuity was observed (red arrow) with tumor cells oriented towards the lumen of the vessel. In some vessels, the percentage of vascular damage may result in higher levels without endothelium, composing and forming a palisade-like structure. Shrinkage of the tumor mass observed after oral administration of RES529 resulted in dramatic vessel damage with breaks. PAS positive proteins envelope the damaged vessels in order to hinder a possible bleeding. Combination strategies with anti-angiogenetic compounds and RES529 are necessary to reduce both angiogenesis and vasculomimicry with low number of angiogenetic and vasculogenetic vessels when compared with monotherapy alone. Wide non-vascularized areas were found in RES529/bevacizumab treated tumors supporting a possible fibrotic reaction. Wide areas rich in collagen fibers (stained in blue) indicate the presence of massive fibrosis (row 4, Figure 6).

### 3.7. Orthotopic Intracranial Model with Luciferase U87MG Cells

Treatments and schedule were schematized in Figure 7A. Treatment with 100 mg/kg/day RES529 increased the time necessary to evidentiate an intra-brain tumor lesion (DFS) from 27.0 ± 6.32 to 40.0 ± 21.06 days) whereas bevacizumab showed a DFS = 32.5 ± 5.89 days and sunitinib a DFS 56.50 ± 12.48. The differences were statistically significant only for RES529 and sunitinib (Figure 7B and Appendix A). A dramatic and statistically significant improvement of DFS was observed when RES529 was administered in combination with bevacizumab (DFS = 99.50 ± 80.53 days). The DSF was significant when compared to single treatments. The contribution of RES529 to sunitinib effectiveness was justified by the increment of DFS which arrive to 81.00 ± 33.15 days. Kaplan-Meyer curves for DFS data (Figure 7C and Appendix A) indicated that the combination between RES529 and bevacizumab or between RES529 and sunitinib were significantly more effective than the treatment with both drugs taken individually with hazard ratios of 4.81 and 4.39 versus CTRL in bevacizumab or sunitinib contained combination, respectively. Comparison with single treatments revealed a statistically significant increment in HR for both combinations with HR = 2.37 and HR = 4.74 with RES529 + bevacizumab versus RES529 and bevacizumab, respectively (Appendix A). Similarly, we show in comparison, HR = 2.7 and HR = 2.24 RES529 + sunitinib versus RES529 and sunitinib, respectively. Similar results were obtained with OS in which RES529 displayed an incremental change to 89.5 ± 24.2 days from 50.40 ± 17.64 days in the control (Figure 7D). Bevacizumab showed an OS = 79.2 ± 21.64 days statistically significant in comparison to untreated animals (OS = 42.9 ± 2.7 days) but not different for RES529 (Appendix A). Sunitinib showed an OS = 103.40 ± 29.19 statistically significant only versus CTRL. Improvement in OS showed greater relevancy and statistical significance when RES529 was combined with sunitinib (OS = 162.60 ± 38.94 days) whereas the combination with bevacizumab showed an OS = 152.70 ± 54.28. The increment in OS was statistically significant when the combinations were compared with the monotherapy (Appendix A). Figure 7D show the Kaplan-Meyer curves for the overall survival data with statistical analyses in Appendix A. Comparison of the hazard ratios of single and combined treatments confirms further the improvement in OS of RES529 with HR = 3.03 when tested alone versus CTRL or in combination with bevacizumab (HR = 5.87 versus CTRL, 4.01 versus RES529, and 5.03 versus bevacizumab monotherapies or sunitinib (HR = 4.98 versus CTRL; 4.83 versus RES529, and 3.56 versus sunitinib monotherapies). These results suggest that the combination of RES529 with bevacizumab or sunitinib showed a cumulative effect on DFS and OS parameter in the differentiated U87MG cell model.

### 3.8. Orthotopic Intracranial Model with Tumor Propagating Cells

Since TORC2 regulates the growth and invasiveness of tumor-propagating cells [52,53,54], luciferase-transfected glioma stem cells (GSCs.5 cell line) were injected into mouse brains (Figure 8 and Appendix A). Fifty days were necessary to observe a bioluminescent lesion in the control brains (50.40 ± 17.64 days). Treatments with RES529 (89.50 ± 24.20), bevacizumab (79.20 ± 21.62 days), and sunitinib (103.40 ± 29.19 days) significantly increased the DFS period (+78% for RES529; +57% for BEVA, and +105% for Suni) when compared to controls (Figure 8A and Appendix A). The combination with RES529 increases DFS of about 3-fold (152.70 ± 54.28 days) and 4-fold (162.60 ± 38.94 days) for BEVA and Suni, respectively. Kaplan-Meyer curves showed no significant difference amongst single monotherapies while the association between RES529 and bevacizumab displayed higher efficacy than the treatment with both drugs taken individually (HR = 4.24 for the combination RES + BEVA versus RES529 or HR = 4.83 versus BEVA alone, Figure 8B and Appendix A). Similar results were obtained for the association with Suni with HR = 5.22 versus RES529 and HR = 3.89 versus Suni alone. Increased tumor appearance time was associated with a significant increase in OS (Figure 8B,C and Appendix A). The analysis of Kaplan-Meyer curves indicates that bevacizumab (OS = 103.00 ± 24.1 days) and RES529 (123.25 ± 27.03 days) effectively increased OS of GSCs-5 bearing animals. RES529 increased the effectiveness of bevacizumab both in terms of mean OS (200.70 ± 39.67 days) and reduction of mice in progression (HR = 4.21).

This combination shows significant effectiveness when compared to the administration of RES529 alone (HR = 5.25). For the analysis of OS, sunitinib was additive in the increase of OS with OS = 215.55 ± 33.33 and HR = 7.78. These values show significant increment in comparison to RES529 alone (HR = 5.90) and sunitinib (HR = 2.64) alone.

## 4. Discussion

It has been widely demonstrated that Akt activation modulates the conversion of anaplastic astrocytoma to GBM [27,28,29,30,31,32]. Recently, activation of PIK3CA and Akt pathway members has been shown to be associated with reduced patient survival times [27,31,32]. Activation of PI3K/Akt/mTOR pathways have been found in the majority of patients with rGBM [28].

Loss of function mutations, chromosomal deletions, or epigenetic silencing of PTEN have been found in approximately 40% of GBM cases and have been shown to reduce overall survival of patients [32,55,56]. Activation of several growth factor receptors guarantees elevated levels of activated PI3K/Akt/mTOR and MEK/Erk pathways. Several inhibitors of PI3K/Akt/mTOR pathways have been developed for the treatment of different cancers including GBM. Clinical trials using rapamycin analogs were found to display insufficient anti-tumor activity in patients with rGBM [48,57,58,59]. A possibility could be the heterogeneity of GBM, with its redundant signaling inputs and ability to bypass blockade of individual molecules through compensatory feedback loops. A major hurdle to inhibiting the PI3K pathway in the brain is the BBB, which excludes many therapeutic compounds and thus makes GBM treatment more difficult [44,45,46]. RES529 a dual TORC1/TORC2 small molecule inhibitor is however orally bioavailable and able to cross BBB [44] and lacking affinity for ABCB1/ABCG2. Although antiangiogenetic agents improve treatment of rGBM, BEVA induces also tumor relapse. BEVA is responsible for vascular regression, tumor growth inhibition, and prolonged survival suggesting that this agent shows antitumor effects. There are evidences on direct antitumor effects of BEVA on established GBM and glioma stem-like cells [5,57,58,59,60] and BEVA shows significant anti-tumor potential as monotherapy [61]. It has also been demonstrated that BEVA monotherapy was associated with increased in intra-tumor AKT phosphorylation levels [62]. It may be that higher levels of active MET [59,60,61,62,63] detected in the post-BEVA treatment tumors are responsible for focal and multifocal invasive phenotype [62]. In addition, Glioma stem-like cells (GSC) have been shown to infiltrate and proliferate after post-treatment in hypoxic niches. These cells are the glioma initiating cells (GIC) which are responsible for disease initiation, recurrence, and therapeutic resistance [63]. BEVA transiently normalizes the abnormal structure and function of tumor blood vessels [7,13,28], consequently facilitate enhanced delivery of different compounds, and eventually ameliorate the response to monotherapies. It has been demonstrated that pharmacological inhibition of mTORC1 has minimally effective in GBMs. Some independent reports suggest that dual TORC kinase inhibitors may be more active in GBMs. RES529 is a dual TORC1/TORC2 dissociative inhibitor of the mammalian target of rapamycin protein complex (mTORC) [41,42,43,44]. This agent is the first mTOR targeting drug lacking affinity for ABCB1/ABCG2 [44] and thus having good brain penetration. In this study, the efficacy of RES529 in vitro by using established GBM cell lines and GICs and in vivo by using subcutaneous xenografts and orthotopic intra-brain cell inoculation was used to evaluate the efficacy of RES529 in the intracranial xenograft models alone or in association with anti-angiogenetic compounds.

The in vivo models described here are designed to receive a short-term therapy, 35 days administration of RES529. The subcutaneous model allows us to make the molecular analyses (biochemical and histochemical evaluation) on tumors, harvested at the end of the experiment, subjected to continuous pharmacological pressure (happening throughout the entire length of the experiment). In addition, this model allows calculation of Time to Tumor Progression (TTP) data as well as the percentage of tumors in progression (important parameters for a preclinical study with inroads to clinical development) through volumetric tumor variation over time. Differently, the intracranial model was used to obtain data on “survival” endpoints. However, in this model, mice remained without treatment for several weeks or months after 35 days of consecutive administration of RES529. As such, no enzymatic or histochemical analyses can be performed under continuous pharmacological pressure.

RES529 reduced the percentage of cells in G0/G1 at IC50 values with increments of subG1 cell and G2/M percentage. This was associated to apoptosis in agreement with data from APOSTRAND™ ELISA kit and caspase-3 activity. RES529 induced G2/M cell cycle accumulation and apoptosis in agreement with previous studies [41,42,64]. RES529 treatment alone showed marked decrease in tumor growth (subcutaneous) and increase in survival (orthotopic).

Here, we also report the increased efficacy of RES529 in combination with antiangiogenetic compounds including bevacizumab or sunitinib. The association of Akt or TORC inhibitors with bevacizumab has been previously reported along with a Phase II study of bevacizumab and temsirolimus combination therapy for recurrent GBM [65]. Although sunitinib is not the standard of care in the treatment of GBM patients, anti-angiogenetic sunitinib was used as the experimental reference compound showing mechanism of action (MOA) with respect to bevacizumab. Here, we determined the sensitizing effects of the small molecule and oral available dual TORC1/TORC2 inhibitor RES529 in combination with an anti-VEGF blocking antibody (bevacizumab) and a small molecule tyrosine kinase inhibitor (sunitinib). Notably, sunitinib has been extensively used in GBM preclinical studies attesting its use as an anti-angiogenetic compound in this experimental setting.

In addition, RES529 had a more than additive sensitizing activity versus antiangiogenetic therapies as in bevacizumab or sunitinib probably through the inhibition of alternative pathways to VEGFR1.

## 5. Conclusions

In summary, our data indicate a complex network of interactions between angiogenesis and TORC2 activity. RES529, the first dual TORC1/TORC2 dissociative inhibitor, lacking affinity for ABCB1/ABCG2 and having good brain penetration. RES529 was shown to be active in GBM preclinical/murine models. As such, results shown here support the use of dual administration of RES529 with antiangiogenic agents as a GBM therapeutic modality. Our results are suggestive for RES529 as a novel tool for tackling GBM giving credence to its use in clinical trial for patients with GBM treated in association with anti-angiogenetic compounds.

## Figures and Tables

**Figure 1 cancers-11-01604-f001:** Constitutive activation and inhibitory activities of TORC1/TORC2 inhibitor (RES529) on phosphatidylinositol 3-kinase (PI3K)/Akt/mTOR in 11 glioblastoma multiforme (GBM) cell lines present in our cell cohort (is lacking the sole SF268 cell line which was not examined). Analyses was performed in whole cell lines cultured in 96-well microplates at 20,000 cells/wells. Untreated human brain microvascular endothelial (hBMVEC) cells were used as negative controls whereas vascular endothelial growth factor (VEGF) triggered hBMVEC were used as positive controls. (**A**) ELISA determination was assessed by using p-Ser473 Akt, p-Thr308 Akt, p-Ser2448 mTOR, and p-Ser 37/46 4E-BP1 antibodies on 11 GBM cell lines of our cell line cohort. Assays were performed in triplicate in a semi-quantitative manner by using arbitrary fluorimetric units. A red line indicates the cutoff of ELISA. Statistics: * *p* < 0.05 for the comparison between basal hBMVEC cells (negative control) and VEGF-treated cells (positive control) for all enzymatic activities. Next, GBM cell lines were compared with untreated (−VEGF) hBMVEC. These cells showed significant higher enzymatic activities compared to untreated hBMVEC (* *p* < 0.05). * *p* < 0.05 was added to indicate the statistically higher enzymatic values in the GBM cell lines compared to untreated (VEGF) hBMVEC. (**B**) Enzymatic inhibition by RES529 was evaluated by ELISA in U87MG and CSCs-5 cells measured through the decrease of p-Ser473 Akt and pSer235/23-S6 expression levels versus basal levels as percentage of inhibition in U87MG and CSCs5 cells. Assays were performed in triplicate. (**C**) Antiproliferative effects of RES529 on established glioblastoma cell lines and patient derived Glioma Initiating cells (GICs) with IC50 values evaluated in 13 GBM cell lines and seven GICs. In order to verify if normal endothelial cells were sensitive to RES529, we used the hBMVEC untreated or stimulated with VEGF (10 ng/mL). (**D**) Cell cycle phase distribution in U87MG, U251, A172, and T98G. Comparison between untreated cells and cultures treated with RES529 at relative IC20 value.

**Figure 2 cancers-11-01604-f002:** In vivo experiments: RES529 inhibition of tumor growth of U87MG cells subcutaneously injected in female nu/nu mice (xenograft model). Each group is represented by five mice with two tumors in the flank. (**A**) Schematic protocol for the dose-dependent experiment with doses of 25, 50, and 100 mg/kg/day, (**B**) tumor weight analyzed at the end of treatments. (**C**) Time to Progression (TTP, days) and (**D**) percentage of mice in progression plotted for the time (Kaplan-Meyer curves) for the U87MG cell model. Statistical analyses are shown in Appendix A.

**Figure 3 cancers-11-01604-f003:** In vivo experiments: RES529 modified tumor growth of luciferase transfected U87MG cells orthotopically injected in the brain of female nu/nu mice. Ten mice per group with two tumors each were evaluated. (**A**) Schematic protocol for the dose-dependent experiment with doses of 25, 50, and 100 mg/kg/day. (**B**) Disease-free survival values. (**C**) Percentage of mice in progression plotted for time (Kaplan-Meyer curves). (**D**) Overall survival values. (**E**) Percentage of dead mice in time (Kaplan-Meyer analysis). Statistical analyses are shown in Appendix A.

**Figure 4 cancers-11-01604-f004:** Anti-angiogenetic effects of RES529 in combination with bevacizumab (Beva). (**A**) Tubule formation of hBMVECs cultured with VEGF 10 ng/mL with added RES at 1.0 and 5.0 μM, bevacizumab, or combination RES529 (1.0 μM) and bevacizumab (qualitative images and branching index analysis). Magnification 50×. (**B**) IC50 values calculated for the anti-proliferative effects (5.02 ± 1.53 μM) of RES529 on endothelial cells and tubule formation (IC50 (mM) = 8.23 ± 3.37). (**C**) Matrigel plugs, containing saline (vehicle) or VEGF 10 ng/mL, harvested in animals untreated or treated with 50 mg/kg/day (one week), bevacizumab 4 mg/kg/day (one administration at day 1) or combination. (**D**) Analyses of vessel count per microscopic field at 20× and hemoglobin content. For each assay, *n* = 5 (five replicates). Data were presented as mean ± standard deviation (SD). Statistics: (**A**) * *p* < 0.05 for comparison VEGF versus CTRL; (**D**) * *p* < 0.05 for comparison between VEGF and CTRL; ** *p* < 0.05 for the comparison between VEGF and RES529 (RES), bevacizumab (Beva) and RES + Beva; finally # *p* < 0.05 for the comparison between RES and RES + Beva. Experiments in this figure were repeated three times, and each time the similar results were obtained.

**Figure 5 cancers-11-01604-f005:** In vivo experiments: RES529 modified tumor growth of U87MGcells subcutaneously injected in female nu/nu mice (xenograft model). Each group is represented by five mice with two tumors in the flank. (**A**) Schematic protocol for the combination experiment (Table 1) with 50 mg RES529 and/or BEVA (4 mg/kg iv every four days) or sunitinib (40 mg/kg per overall survival (OS)); (**B**) tumor weight analyzed at the end of treatments. (**C**) Time to Progression (TTP, days). (**D**) Kaplan Meier curves generated for percentage of tumors in progression Statistical analyses are shown in Appendix A.

**Figure 6 cancers-11-01604-f006:** Histological and histochemical analyses on U87MG xenografts. RES529 (50 mg/kg os qd) and bevacizumab (4 mg/kg iv every 14 days) were administered alone or in combination in U87MG xenografted mice to assess modification in the vessel network and vasculogenic mimicry. The first row shows CD34 expression in untreated U87MG tumors as well as the vascular network after Periodic Acid-Schiff (PAS) and Trichrome Masson staining. The second row shows U87MG tumors treated with bevacizumab. The third row shows U87MG tumors treated with RES529 and the fourth row shows U87MG tumors treated with combination of RES and bevacizumab. Magnification: 1 cm = 100 µm.

**Figure 7 cancers-11-01604-f007:** In vivo experiments: RES529 inhibition of tumor growth of luciferase transfected U87MG cells orthotopically injected in the brain of female nu/nu mice. Each experimental group included 10 animals. (**A**) Schematic protocol for the dose-dependent experiment with doses with 50 mg/kg/day RES529 and/or BEVA (4 mg/kg iv every four days) or sunitinib (40 mg/kg per OS). (**B**) Disease-free survival (DFS) values. (**C**) Percentage of mice in progression plotted for time (Kaplan-Meyer curves). (**D**) Overall survival (OS) values. (**E**) Percentage of dead mice in time (Kaplan-Meyer curves). Statistical analyses are shown in Appendix A.

**Figure 8 cancers-11-01604-f008:** In vivo experiments: RES529 inhibition of tumor growth of luciferase transfected GSCs-5 cells orthotopically injected in the brain of female nu/nu mice. Each experimental group included 10 animals. (**A**) Disease-free survival (DFS) values. (**B**) Percentage of mice in progression plotted for time (Kaplan-Meyer curves). (**C**) Overall survival (OS) values. (**D**) Percentage of dead mice (Kaplan-Meyer curves). Statistical analyses are shown in Appendix A.

**Table 1 cancers-11-01604-t001:** Experimental in vivo procedures.

Experiment #	Experimental Model	Injected Cells	Treatments and Groups	Mice for Group	Experimental Measures and Endpoints
01	Subcutaneous xenograft	U87MG 1 × 10^6^/tumor	RES529 group (25 mg/kg/day)RES529 group (50 mg/kg/day)RES529 group (100 mg/kg/day)	5 animals/group(2 tumors/animal)	-Tumor volume;-Tumor weight-TTP-Tumors in progression (%)-Biochemical evaluations
02	Subcutaneous xenograft	A172 1 × 10^6^/tumor	RES529 group (25 mg/kg/day)RES529 group (50 mg/kg/day)RES529 group (100 mg/kg/day)	5 animals/group2 tumors/animal	-Tumor volume;-Tumor weight-TTP-Tumors in progression (%)-Biochemical evaluations
03	Subcutaneous xenograft	T98G 1 × 10^6^/tumor	RES529 group (25 mg/kg/day)RES529 group (50 mg/kg/day)RES529 group (100 mg/kg/day)	5 animals/group2 tumors/animal	-Tumor volume;-Tumor weight-TTP-Tumors in progression (%)-Biochemical evaluations
04	Orthotopic Intracranial xenograft	U87MG 1 × 10^3^	RES529 group (25 mg/kg/day)RES529 group (50 mg/kg/day)RES529 group (100 mg/kg/day)	10 animals/group	-Bioluminescence,-Overall survival-Disease Free Disease (DSF)-Health State-Animal death or distress signs
05	Subcutaneous xenograft	U87MG 1 × 10^6^/tumor	RES529 (50 mg/kg every day); Bevacizumab (4 mg/kg iv every 14 days) Sunitinib (40 mg/kg po qd)RES529 + BevacizumabRES529 + Sunitinib	5 animals/group2 tumors/animal	-Tumor volume;-Tumor weight-TTP-Tumors in progression (%)-Biochemical evaluations
06	Subcutaneous xenograft	A172 1 × 10^6^/tumor	RES529 (50 mg/kg every day); Bevacizumab (4 mg/kg iv every 14 days) Sunitinib (40 mg/kg po qd)RES529 + BevacizumabRES529 + Sunitinib	5 animals/group2 tumors/animal	-Tumor volume;-Tumor weight-TTP-Tumors in progression (%)-Biochemical evaluations
07	Orthotopic Intracranial xenograft	U87MG 1 × 10^3^	RES529 (100 mg/kg every day); Bevacizumab (4 mg/kg iv every 14 days) Sunitinib (40 mg/kg po qd)RES529 + BevacizumabRES529 + Sunitinib	10 animals/group	-Bioluminescence,-Overall survival-Disease Free Disease (DSF)-Health State-Animal death or distress signs
08	Orthotopic Intracranial xenograft	GSCs-5 1 × 10^3^	RES529 (100 mg/kg every day); Bevacizumab (4 mg/kg iv every 14 days) Sunitinib (40 mg/kg po qd)RES529 + BevacizumabRES529 + Sunitinib	10 animals/group	-Bioluminescence,-Overall survival-Disease Free Disease (DSF)-Health State-Animal death or distress signs

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
