# Peer review of "The Brain Penetrating and Dual TORC1/TORC2 Inhibitor, RES529, Elicits Anti-Glioma Activity and Enhances the Therapeutic Effects of Anti-Angiogenetic Compounds in Preclinical Murine Models"

_cancers, 2019, doi:10.3390/cancers11101604_

Round 1
Reviewer 1 Report
The work by Gravina et al. nicely describes the therapeutic effect of RES529 in a range of GBM models. The work is novel and interesting and the authors have comprehensively looked at it’s function in different models. I have a few concerns that need to be addressed.
1. In section 2.5. Glioblastoma xenograft model. The authors need to specify how the treatment was administered.
2. In line 159 the authors need to provide the reference.
3. How was these tumours measured in section 2.5.
4. In section 2.7. Orthotropic intra-brain model, two experiments are described, the authors need to separate them for ease of understanding. Also they need to describe how tumour cells were injected is this model is this an intracranial model?
5. Result section Figure 1A, the authors indicate that there is high expression of p-Ser473 Akt, p-Thr408 Akt, p-Ser2448 mTOR and Thr46/47-4E-BP1 in GBM models however they don’t indicate what this have been compared to in terms of controls? What are the control used in this section?
6. In Figure 2 again there need to be controls included. The Authors also need to justify the selection of the cell lines for Figure 2.
7. A172 data is not reported in Figure S1, same for Figure S2.
8. Sentence 284-286 is grammatically wrong.
9. 332 is VEGF instead of VEG
10. The Authors should include the number of animals in each experiment in the figure legends.
Author Response
The work by Gravina et al. nicely describes the therapeutic effect of RES529 in a range of GBM models. The work is novel and interesting and the authors have comprehensively looked at its function in different models. I have a few concerns that need to be addressed.
Question 1. In section 2.5. Glioblastoma xenograft model. The authors need to specify how the treatment was administered.
First of all we have to thank this reviewer for his/her overall good opinion on our report.
We have better clarify as treatments were made accordingly. In particular:
We specified the follow treatment schedule:
(1) Control (vehicle);(2) RES529 (25 mg mg/kg/5 Day/week, PO); (3) RES529 (50 mg mg/kg/5 Day/week, PO); (4) RES529 (100 mg mg/kg/5 Day/week, PO).
similarly was specified also the follow treatment schedule:
(1) Control (vehicle); (2) Bevacizumab 4 mg/Kg i.p. every 14 days; (3) RES-529 (50 mg mg/kg/5 Day/week, PO); (4) BEV plus RES-529; (5) Sunitinib (40 mg/Kg PO qd); (6) SUNI plus RES.
So, to better specify we added the definition of the acronyms and therefore PO = per os (oral gavage) and ip = intraperitoneal and qd= quaque die/ "every day."
see new statement at line 165-170.
In line 159 the authors need to provide the reference.Reference was provided at line 156
How was these tumours measured in section 2.5.This was clarified, see lines 175-177
Numerous formulas have been used in the literature to assess volumetric analyses in xenografts. Considering an ovoid-like shape of subcutaneous xenografts, we used the formula 4/3πR1xR2xR3 [16, 18] where R1 is the radius relative to the width, R2 is the radius relative to the length and R3 the one relative to the thickness.
Question 4. In section 2.7. Orthotropic intra-brain model, two experiments are described, the authors need to separate them for ease of understanding.
Also they need to describe how tumour cells were injected is this model is this an intracranial model?
(i) This is clarifed in the new version of the paper with the addition of "Experiment #1" and "Experiment # 2" (highlighted in yellow) at lines 214 and 216
(ii) Yes, these are intracranial models. For detailed procedures of orthotopic intra-brain cell inoculation, we added a reference (reference #51) in addition to our two references 16 and 18.
Baumann BC, Dorsey JF, Benci JL, Joh DY, Kao GD. Stereotactic intracranial implantation and in vivo bioluminescent imaging of tumor xenografts in a mouse model system of glioblastoma multiforme. J Vis Exp. 2012 Sep 25;(67). pii: 4089.
So, reference numbering starting to old reference 51 is increased of 1 unit.
Question 5. Result section Figure 1A. The authors indicate that there is high expression of p-Ser473 Akt, p-Thr408 Akt, p-Ser2448 mTOR and Thr46/47-4E-BP1 in GBM models however they don’t indicate what this have been compared to in terms of controls? What are the control used in this section?
In this figure the authors are evaluating as fluorimetric units obtained from the enzymatic assays from four kinases of the Akt/mTOR pathways in GBM cell lines grown under basal conditions (without growth factors or cytokines). In this condition these enzymatic activities are not completely activated also if different basal activation would be associated with autocrine stimulation. Notably this signaling pathways is activated in presence of GFs (EGF, FGF, VEGF, HGF TGF beta etc.) and cytokines (CXCR4, CXCR1, MCP1 etc.), being Akt/mTOR signals the basis of the differently molecular pathways.
To obtain data from cell-based ELISAs, the authors followed the instructions of manufactures that is: cultured the cells in 96 well microplates, fixed with 4% formalin, permeabilized with 0.1% triton X100, treated with different FITC-antibodies and fluorescence analyzed at a specific reader.
In accordance with data sheets of single ELISAs, the authors generated reactive blanks (BR) and negative controls without fluoro-chrome or cells in manner to subtract from single fluorescence values the values of RFU. In addition authors normalized the fluorimetric units thus obtained with GAPDH used as housekeeping gene. Some of this information was added at lines 165-170.
However if the need to have reference controls is what is required by the reviewer, was added the values of RFU relative to human brain derived normal endothelial cells (hBMVEC) which may represent our negative control when grown in absence of angiogenic stimuli since they are negative for Akt activation. The same endothelial were administered with 10 ng/ml hVEGF (positive control). In this case hBMVEC are activated, trigger a robust Akt activity and form in matrigel tubule structures similar to vessels.
GBM cell lines and hBMVEC treated with VEGF showed significantly higher Akt/mTOR activity versus negative control and cutoff.
We added the use of these cells in MM (lines 142-151 (yellow highlighted statement) and in the result section at lines 250-256. We added in figure 1A a red line representing the cutoff of single ELISAs. Figure legend was modified accordingly.
Question 6a. In Figure 2 again there need to be controls included.
In this figure the authors are analyzing the IC50 values for each cell lines with RES529. The negative control is represented by the 100% of proliferation/viability in absence of drug (0 concentration) for each assay and the percentage of viability was evaluated increasing drug concentration. But this is implicit in the concept of IC50
I do not understand which controls this reviewer is referring to.
Making an understanding effort, we thought that perhaps the reviewer also in this case was referring to the negative and positive controls of normal brain cells! So, also in this case we used the hBMVEC untreated or stimulate with VEGF in order to trigger a robust Akt activity. As it is possible to see in the new figure 2A untreated hBMVEC was not sensitive to RES529 if not at the highest doses with IC50 > 50mM whereas VEGF stimulated hBMVEC resulted sensitive to RES529 with IC50<20 mM.
Text and Figure legend was modified accordingly. see lines 263-266
Question 6b. The Authors also need to justify the selection of the cell lines for Figure 2.
Surely, was inconceivable to use all 11 cell lines for further in vitro and in vivo biological analyses. So, we selected U251, U87MG, A172 and T98G since these cell lines are among the most used in the literature for GBM research. These cells were also widely used in our lab and are also known for their molecular arrangements related to proliferation, autophagy, drug sensitivity and apoptosis including PTEN, Akt activation, expression of growth factor receptors, p53, Rb, MGMT, autophagy modulating molecules etc.. The differential molecular assets (characterizing these GBM cell lines) differentially regulates many pharmacological responses. These cells are also previously used in order to evaluate their sensitivity to PI3K, Akt and mTOR inhibitors and therefore it was possible to make comparisons with these compounds when was used new compounds including RES529. In addition, these cells may be considered moderately sensitive to RES529 and are widely representative for GBM. U87MG was also available in the lab as luciferase transfected cells ready for bioluminescence analysis of intra-brain tumors. similarly we used CSCs-5 glioma stem like cells since we obtained successfully luciferase transfected cell line for in vivo analyses.
Some of these statements were added in the results at lines 268-273
Question 7. A172 data is not reported in Figure S1, same for Figure S2.
This was modified accordingly S1 and S2 represent now the results for A172 and T98G respectively.
Question 8. Sentence 284-286 is grammatically wrong.
This was changed
Question 9. 332 is VEGF instead of VEG
This was changed see figure legends
Question 10. The Authors should include the number of animals in each experiment in the figure legends.
The number of animals for group was 5 with 2 tumors in the flanks (subcutaneous xenografts) and 10 for orthotopic intra-brain tumors.
This was changed see figure legends
Reviewer 2 Report
In this article, Gravina and collaborators showed that an TORC1/TORC2 inhibitor has a potential effect on pre-clinical glioblastoma models, mainly when coupled with anti-angiogenic drugs.
This is an interesting study with a impressive amount of work. However, the amount of experiments, especially in vivo, can make the manuscript unclear.
I recommend several extra-experiments and controls before accepting it.
Major points :
- many graphs in the article were not statistically analyzed. The authors should add statistical tests in these figures : Figure 1A, the whole Figures 2, 5 and 6.
- the authors claimed that RES-529 crosses blood-brain barrier, why did the authors perform subcutaneous experiments ? These figures should be added as supplemental material.
- No immunohistological analyses were done on tumors, especially on GSCs-5 which could bring some interesting information, especially after all combo treatments (IHC of blood vessels...).
- Figure 5C : it is unclear what the authors want to show, please explain.
- Figure 6A : please add a control without any additional growth factors or inhibitors. Is it a vascular mimicry assay or an invasion assay?
- Figure 6C should be removed as not bringing any new information.
- Why did the authors use Sunitinib in their experiments as it is not used in clinics for treating patients with glioblastoma.
Minor points ;
please read carefully the manuscript as many typing mistakes are present.
Author Response
In this article, Gravina and collaborators showed that an TORC1/TORC2 inhibitor has a potential effect on pre-clinical glioblastoma models, mainly when coupled with anti-angiogenic drugs.
This is an interesting study with a impressive amount of work. However, the amount of experiments, especially in vivo, can make the manuscript unclear.
I recommend several extra-experiments and controls before accepting it.
First of all we have to thank this reviewer for his/her overall good opinion of the report.
We have better clarify as treatments were made. In particular:
Question 1. many graphs in the article were not statistically analyzed. The authors should add statistical tests in these figures : Figure 1A, the whole Figures 2, 5 and 6.
We add statistical analyses both in the figure and figure legends.
In figure 1A the authors are showing that GBM cell lines show high levels of Akt/mTOR activation. The text was performed by cell-based Elisa assays in triplicate and values. So for each cell line we show the mean of fluorescence units measured ± the standard deviation (SD) of each cell determination.
We added in the legend of figure 1the following sentences and modified figure accordingly.
" Statistics: p<0.05 for the comparison between basal hBMVEC cells(negative control) and VEGF-treated cells (positive control) for all enzymatic activities. T98G and SW1783 showed higher pSer347activity when compared with those observed for U251, U118, U138, D54, LN229 and LN19. We use the symbol # to indicate p<0.5 for these statistic analyses. LN19 and U138 showed the lower enzymatic pSer347activity when compared with the other cell lines. The * indicates p<0.05 for this statistics. U251and SW1783 showed higher pThr408 activity when compared with those observed for A172, U118, U138, D54, U373 and LN19. We use the symbol # to indicate p<0.5 for these statistic analyses. U118, U138, U373 and LN19 showed the lower enzymatic pSer347activity when compared with the other cell lines. The * indicates p<0.05 for this statistics. SW1783 showed higher pSer2448 mTOR activity when compared with those observed for U251, A172, U118, U138, D54, U373 and LN19. We use the symbol # to indicate p<0.5 for these statistic analyses. U251, U118, U138 LN19 showed the lower enzymatic pSer2448 mTOR activity when compared with the other cell lines. The * indicates p<0.05 for this statistics. U251, U118, U138 LN229 showed the lower enzymatic p-Ser 37/46 4E-BP1 activity when compared with the other cell lines. The * indicates p<0.05 for this statistics."
this and other are shown at lines 285-307
In figure 2 the authors are showing that GBM cell lines show the IC50 values calculated for each cell line used in this paper.
FIGURE 2A:
Statistics: p<0.05 (*) for the comparison between basal hBMVEC cells(negative control) and VEGF-treated cells (positive control). Although this is not the primary purpose, we found that the hBMVEC cells were insensitive to RED-529 and DZ54 cells were the most resistant GBM cells whereas the LN229, BT50EF, LN19, U138 and U87MG cells were the more sensitive . We use the symbol # to indicate p<0.5 for these statistic analyses.
We added in the legend the following sentences and modified figure accordingly.
FIGURE 2D: The comparison of percentage of cells in subG1 cell cycle phase indicates that T98G cells were the high sensitive versus RES-529 being the subG1 cell population at IC50 statistically higher when compared to other cells lines. Differently U251 cells show the lower percentage of subG1 cell population at IC20 value. We added the symbol # to indicate the p<0,05 for these statistical comparisons. * for p<0.05 for specific comparisons.
FIGURE 2F: * p<0,05 vs respective basal level
Figure 2G: * p<0,05 vs basal values
Statistics for figure 2 was added at lines 323-332
Figure 5A: we added * p<0,05 for comparison VEGF vs CTRL
Figure 5D: we added * p<0.05 for comparison between VEGF and CTRL
** P<0,05 VS for the comparisons between VEGF and RES529 (RES), Bevacizumab (Beva) and RES+Beva; finally # p<0,05 RES vs RES+Beva
Statistics for figure 5 was added at lines 435-439
Figure 6 is now figure S3, So, Figure S3B show the following sentence: * p<0,05 vs CTRL and # p<0,05 RES 5.0 mM vs bevacizumab
Statistics for figure S3 was added at lines 689-691
Question 2. the authors claimed that RES-529 crosses blood-brain barrier, why did the authors perform subcutaneous experiments ? These figures should be added as supplemental material.
As indicated in our report we tested RES529 effectiveness both in subcutaneous and intra-brain xenograft models. These two in vivo models were conceived to obtain different information on RES529 administration. Both models are designed to receive a short term therapy (35 day administration with RES529). The subcutaneous model was planned to collect tumors at the end of experiment in manner that tumor cells were subjected to continuous and selective pressure (throughout the entire length of experiment) due to presence of RES529. In this manner, we can collect data of the Time of Tumor Progression (TTP) and the percentage of tumors in progression (important parameters for a preclinical study with inroads in clinical) through the volumetric variation in the time of tumors and harvest tissues for further IHC analysis and biochemical evaluations in this status of continuous and selective pharmacological pressure. The information derived by this model have added a significant body information to the manuscript.
Differently the intra-brain model was used to obtain data on survival endpoint and since the mice remained without treatments for several weeks or months, no information on molecular signature were obtainable from this model conceived in this manner.
This is particularly stressed at lines 99-111
In addition with the intra-brain model, indeed, we want to simulate as a pharmacological treatment could modify the recurrence of a tumor mimicking a possible clinical situation in which a patient with GBM was surgically treated. In this manner a low number of tumor cells, remaining the wound bed, could be induced to re-grow resulting in a recurrence.
This was the central theme of a series of our previous reports (see below) published in important scientific journals. We believe, indeed, that inoculating a small number of cells (3 x103) and starting drug administration when no bioluminescence was detected we could significantly mimic this particular clinical situation with a treatment immediately close to surgery. The alternative could have been surgery but this is an impractical procedure. Animals with tumor survive with different times due different engraftment and animal responses. The authors are aware that IHC and biochemical evaluation could be obtained also by using also the intra-brain model.
REFERENCES:
Gravina GL, Mancini A, Colapietro A, Delle Monache S, Sferra R, Vitale F, Cristiano L, Martellucci S, Marampon F, Mattei V, Beirinckx F, Pujuguet P, Saniere L, Lorenzon G, van der Aar E, Festuccia C. The Small Molecule Ephrin Receptor Inhibitor, GLPG1790, Reduces Renewal Capabilities of Cancer Stem Cells, Showing Anti-Tumour Efficacy on Preclinical Glioblastoma Models. Cancers (Basel). 2019 Mar 13;11(3).
Festuccia C, Gravina GL, Giorgio C, Mancini A, Pellegrini C, Colapietro A, Delle Monache S, Maturo MG, Sferra R, Chiodelli P, Rusnati M, Cantoni A, Castelli R, Vacondio F, Lodola A, Tognolini M. UniPR1331, a small molecule targeting Eph/ephrin interaction, prolongs survival in glioblastoma and potentiates the effect of antiangiogenic therapy in mice. Oncotarget. 2018 May 11;9(36):24347-24363.
Gravina GL, Mancini A, Mattei C, Vitale F, Marampon F, Colapietro A, Rossi G, Ventura L, Vetuschi A, Di Cesare E, Fox JA, Festuccia C. Enhancement of radiosensitivity by the novel anticancer quinolone derivative vosaroxin in preclinical glioblastoma models. Oncotarget. 2017 May 2;8(18):29865-29886
Calgani A, Vignaroli G, Zamperini C, Coniglio F, Festuccia C, Di Cesare E, Gravina GL, Mattei C, Vitale F, Schenone S, Botta M, Angelucci A. Suppression of SRC Signaling Is Effective in Reducing Synergy between Glioblastoma and Stromal Cells. Mol Cancer Ther. 2016 Jul;15(7):1535-44.
Gravina GL, Mancini A, Marampon F, Colapietro A, Delle Monache S, Sferra R, Vitale F, Richardson PJ, Patient L, Burbidge S, Festuccia C. The brain-penetrating CXCR4 antagonist, PRX177561, increases the antitumor effects of bevacizumab and sunitinib in preclinical models of human glioblastoma. J Hematol Oncol. 2017 Jan 5;10(1):5.
Vignaroli G, Iovenitti G, Zamperini C, Coniglio F, Calandro P, Molinari A, Fallacara AL, Sartucci A, Calgani A, Colecchia D, Mancini A, Festuccia C, Dreassi E, Valoti M, Musumeci F, Chiariello M, Angelucci A, Botta M, Schenone S. Prodrugs of Pyrazolo[3,4-d]pyrimidines: From Library Synthesis to Evaluation as Potential Anticancer Agents in an Orthotopic Glioblastoma Model. J Med Chem. 2017 Jul 27;60(14):6305-6320.
Question 2. No immunohistological analyses were done on tumors, especially on GSCs-5 which could bring some interesting information, especially after all combo treatments (IHC of blood vessels).
I agree with this reviewer on the fact that these analyses could bring important information for the vessel formation and function. In this old version of this report, we have not addressed this aspect because we didn't want to crowd the report.
Many aspects of angiogenetic changes after treatment with bevacizumab combined with compounds that alter the Akt / mTOR signaling pathway had been addressed by several authors and by us in two previous our reports [16, 18]. After initial antitumor effects, which "normalize" angiogenetic network, Bevacizumab, indeed, was demonstrated to induce protective pathways able to increase vasculogenesis which takes place to angiogenesis. We observed, for example, that the CXCR4/SDF1 axis [Gravina et al JHO 2017, ref 16] was activates and this mechanism induced the recruitment of circulating monocytes, murine endothelial cell precursors and pericytes. Similarly the activation of the Ephrin receptor family members (mainly EphA2) was able to regulate cancer stem cell recruitment and vasculomimicry [Festuccia et al Oncotarget 2018, ref 18].
So, we add the new figure 7 in which CD34 expression (murine and human), PAS and trichromic staining were shown to demonstrate the reduction of angiogenesis (murine CD34 positive vessels) after all treatments as well as increased vasculogenesis as vessel structure sustained by human CD34 GBM cells in bevacizumab U87MG xenografts. Both mechanisms were reduced in combined treatments with RES-529 and bevacizumab (see new paragraph 3.5 at line 491-519)
For CD34 staining we used purified anti-mouse CD34 Antibody (MEC14.7) and anti human CD34 antibody (clone 581) which were purchased from Biolegend London United Kingdom (this specification was included also in the material and methods at lines 207-209).
We show the analysis performed on U87MG cells grown as xenografts and not on the CSCs-5 xenografts or orthotopic tissues for the reasons given in the previous answer. Among other things, CSC-s5 (like most stem cells) do not grow subcutaneously. Nevertheless U87MG are a good model to study angiogenesis/vasculogenesis as may indicate the numerous articles (<280) which are identified in PubMed.
Question 3. - Figure 5C : it is unclear what the authors want to show, please explain.
In the figure 5C we analyzed the induction of vascular structures by VEGF in matrigel plugs inoculated in the nude mice. We observed that the natural blocking compound of endothelia cell growth and differentiation (bevacizumab) reduced significantly the number and size of vascular structures. RES529 reduced vessels formation and showed synergistic effects with bevacizumab. We better explain this aim in the text (lines 415-422). We added also method for matrigel plug since this was lacking in the old version of paper (see method lines 228-237
Question 4: Why did the authors use Sunitinib in their experiments as it is not used in clinics for treating patients with glioblastoma.
We agree with the reviewer that sunitinib was not a standard of care in the treatment of GBM patients although a number of clinical trials (please, see the reference quoted below) used this anti-angiogenetic agent as experimental compound. However, the reasons for using sunitinib in this report was mainly based on the specific mechanism of action that this inhibitor has with respect to bevacizumab. in this regard we determined if the sensitizing effects of the small molecule and oral available dual TORC1/TORC2 inhibitor, RES529 was more evident in combination with an anti-VEGF blocking antibody (bevacizumab) or a tyrosine kinase inhibitor (sunitinib, reference compound). Anyway, sunitinib was extensively used in GBM preclinical studies attesting the interest towards this anti-angiogenetic compound in experimental setting
This was better specified at lines 643-652
REFERENCES
1: Balaña C, Gil MJ, Perez P, Reynes G, Gallego O, Ribalta T, Capellades J, Gonzalez S, Verger E. Sunitinib administered prior to radiotherapy in patients with non-resectable glioblastoma: results of a phase II study. Target Oncol. 2014 Dec;9(4):321-9.
2: Hutterer M, Nowosielski M, Haybaeck J, Embacher S, Stockhammer F, Gotwald T, Holzner B, Capper D, Preusser M, Marosi C, Oberndorfer S, Moik M, Buchroithner J, Seiz M, Tuettenberg J, Herrlinger U, Wick A, Vajkoczy P, Stockhammer G. A single-arm phase II Austrian/German multicenter trial on continuous daily sunitinib in primary glioblastoma at first recurrence (SURGE 01-07). Neuro Oncol. 2014 Jan;16(1):92-102.
3: Kreisl TN, Smith P, Sul J, Salgado C, Iwamoto FM, Shih JH, Fine HA. Continuous daily sunitinib for recurrent glioblastoma. J Neurooncol. 2013 Jan;111(1):41-8.
4: Pan E, Yu D, Yue B, Potthast L, Chowdhary S, Smith P, Chamberlain M. A prospective phase II single-institution trial of sunitinib for recurrent malignant glioma. J Neurooncol. 2012; 110:111-8.
5: Reardon DA, Vredenburgh JJ, Coan A, Desjardins A, Peters KB, Gururangan S, Sathornsumetee S, Rich JN, Herndon JE, Friedman HS. Phase I study of sunitinib and irinotecan for patients with recurrent malignant glioma. J Neurooncol. 2011 Dec;105(3):621-7.
6: Scott BJ, Quant EC, McNamara MB, Ryg PA, Batchelor TT, Wen PY. Bevacizumab salvage therapy following progression in high-grade glioma patients treated with VEGF receptor tyrosine kinase inhibitors. Neuro Oncol. 2010 Jun;12(6):603-7
Question 5 - Figure 6A : please add a control without any additional growth factors or inhibitors. Is it a vascular mimicry assay or an invasion assay
When require was made with the addition of a negative control without stimuli. Figure 6 is now figure S3.
YES it is vascular mimicry)
Question 7 - Figure 6C should be removed as not bringing any new information.
OK this was made
Minor points ;
please read carefully the manuscript as many typing mistakes are present.
This was made
Reviewer 3 Report
The manuscript as presented by the authors is difficult to read because it contains a lot of errors. In Figure 8 and 9 for example the panels F,G, H and I are missing. The axis labeling is often wrong (Kaplan Meier analysis, Time to progression?). These are just examples. The manuscript is written in a very strange style, basically the whole manuscript reads like a methods section and the reviewer feels that it has been written last minute. Because of these many errors the reviewer can at this point not provide an in depth review of the manuscript. The authors should rewrite the manuscript, include all figure panels, and fix the errors before submission.
Author Response
The manuscript as presented by the authors is difficult to read because it contains a lot of errors. In Figure 8 and 9 for example the panels F,G, H and I are missing. The axis labeling is often wrong (Kaplan Meier analysis, Time to progression?). These are just examples. The manuscript is written in a very strange style, basically the whole manuscript reads like a methods section and the reviewer feels that it has been written last minute. Because of these many errors the reviewer can at this point not provide an in depth review of the manuscript. The authors should rewrite the manuscript, include all figure panels, and fix the errors before submission.
I was very baffled by the fact that this reviewer did not want to do a review.
Nevertheless, the authors they rolled up their sleeves and began to answer the questions of the reviewers including thus review. We have largely modified the text and fixed the figures as requested by this reviewer:
figure 3 lines 359-365
figure 4 lines 387-392
legend new figure 6, lines 484-490
figure 8 lines 584-591
figure 9 lines 625-630
The work also improved based on the observations of the other two reviewers who that with their work have shown respect of our research group. Surely there were a lot of errors as indicated but as such they did not prevent us from reading and understanding our work.
Round 2
Reviewer 1 Report
The Author have made an effort to reply to most of my concerns including the addition of positive and negative controls, however some of the new comments and sentences to the manuscript are confusing. There is a need for the authors to focus on delivering a clear message and this need to be considered in each section.
Introduction need further improvement the authors have added new section with extra information on how the models have been set-up but this section remains quite confusing. I would suggest stepwise introduction of the models and a reason for the use of each model. The Glioblastoma Xenograft and the orthotopic intra-cranial models described in Methods is still confusing and the use of each model is not clear. I would suggest a table including the type of experiment (Intracranial, subcut, etc) type of cell injected (U87MG, etc), Type of treatment, number of mice, experimental measures and endpoint. In figure 1 the authors need to clarify what they have used in their comparison? Is this comparing the GBM cell lines with the controls +ve and -ve? The comparison in Figure 2 is not clear. Line 320-328 have result/discussion point and does not belong to the figure legend. Line 323 and 328 is not clear. It might be a good idea to combine some figures with the same message including figure 1 and 2, and only display what is needed and move unnecessary data to supplementary figures to make the manuscript easier to understand. Figure 5C resolution is very low. In Figure S3a What was the concentration of RES529 in RE529+Bev treatment, why isn’t this combination used in the other experiments including migration assay as it seems to have the greatest effect. Why did the authors present A172 data in Figure 6 followed by U87MG data in Figure 7?! Figure 6 either need to be changed with Figure S4 or they need to show the data for A172 in Figure 7. The manuscript need grammatical and English language correction as well as correction to the style (bold/un-bold) is used in various sections. There is several errors including the use of comma (,) after full stop (.) and open parenthesis with no closing. There is also a general inconsistency with the way figures are referred to. Figure 3, 4, 6, 8 and 9 there is no need for the tables these could be explained in the legend and illustrated on the graphs. Line 220 the injected number of cells need to be fixed I believe the authors meant 10^3 not 103! Clarify Line 280, antibodies on 11 our 13 GBM cell lines. The resolution of Figure 1, 2 and 7 is very low. Mention RES-529 (RES529) in the same form throughout the document.Author Response
The Author have made an effort to reply to most of my concerns including the addition of positive and negative controls, however some of the new comments and sentences to the manuscript are confusing. There is a need for the authors to focus on delivering a clear message and this need to be considered in each section.
Question 1. Introduction need further improvement the authors have added new section with extra information on how the models have been set-up but this section remains quite confusing. I would suggest stepwise introduction of the models and a reason for the use of each model.
We agree with this reviewer. We extensively rephrased this paragraph and moved it in the discussion section at lines. We hope that this new version, would result clearer.
Question 2. The Glioblastoma Xenograft and the orthotopic intra-cranial models described in Methods is still confusing and the use of each model is not clear. I would suggest a table including the type of experiment (Intracranial, subcut, etc) type of cell injected (U87MG, etc), Type of treatment, number of mice, experimental measures and endpoint.
I added a table as supplemental data (table SI), accordingly.
Question 3. In figure 1 the authors need to clarify what they have used in their comparison? Is this comparing the GBM cell lines with the controls +ve and -ve?
As requested in question 6 figures 1 and 2 are fused to generate new figure 1 (panels A,B of old figure 1 and panels A and E of old figure 2) and Figure S1 (panels C, D, F and G of old figure 2). Replies to the comments are new referred to the new figure numbering.
We thank the reviewer for this request. In order to reply for a question relative to controls addressed by this review in the round 1 of revision, we have added some comparisons for GBM cell lines. We modified the figure 1 and the figure legend accordingly. Statistical comparison was modified accordingly as shown below:
The statistical analysis was performed versus control negative (hBMVEC without VEGF).
I hope this change will provide clarity to figure 1A and 1C. We thank the reviewer for this request. Here, we have provided comparison between each GBM cells with untreated (-VEGF) HBMVECs.
"Statistics: (Figure 1A) p<0.05 (*) for the comparison between basal hBMVECs (negative control) (Figure 1C) ) p<0.05 (*) for the comparison between basal hBMVECs (negative control) and VEGF-treated cells (positive control). * p<0.05 indicated the statistical levels of comparison with untreated (-VEGF) hBMVECs; Untreated (-VEGF) hBMVECs were practically insensitive to RES529 with IC50 > 50 mM and DZ54 cells were the most resistant GBM cells with IC50 of 30 mM. The remaining cell lines showed IC50 values < 20 mM significantly lower when compared to untreated (-VEGF) hBMVECs. D54 cells, however, showed greater statistical significant to these control cells.. For panel Figure S1C, comparison was performed comparing the values at IC20 or IC50 versus respective untreated basal controls. (Figure S1B) comparison of the percentage of cells in subG1 cell cycle phase indicates that cell death was dependent on RES529. The concentration of the subG1 cell population percentage displayed a significantly higher IC50 when compared to IC20 values (* p<0.05); (Figure S1C) caspase 3 activity evaluated at IC20 and IC50 in U87MG, U251, A172 and T98G cells * p<0,05 vs respective basal values; (Figure S1D) caspase 3 activity with or without z-VAD-fmk (pan-caspase inhibitor, 10 μM), necrostatin-1 (necroptosis inhibitor, 50 μM) and ferrostatin-1 (ferroptosis inhibitor, 5 μM). * p<0,05 vs respective basal values."
Question 4. The comparison in Figure 2 is not clear.
see above
Question 5. Line 320-328 have result/discussion point and does not belong to the figure legend. Line 323 and 328 is not clear.
This was made
Question 6. It might be a good idea to combine some figures with the same message including figure 1 and 2, and only display what is needed and move unnecessary data to supplementary figures to make the manuscript easier to understand.
This was done. Figure 1 and 2 are fused in a new figure 01. A new figure S1 was moved to supplementary data. This figure combines panels eliminated during the fusion of the old figure 01 with figure 02.
Question 7. Figure 5C resolution is very low.
We upload images at 300 dpi.
Question 8. In Figure 5a What was the concentration of RES529 in RE529+Bev treatment, why isn’t this combination used in the other experiments including migration assay as it seems to have the greatest effect.
Figure 5a is now figure 4a.
8a. RES529 was added at 0.5 and 5.0 mM when administered alone. This was added accordingly.
8b. We show also the association between 0.5 mM RES529 and 100 ng/ml bevacizumab. These data were not inserted in the old version of this figure. As such, the new figure 4 was prepared accordingly and concentrations indicated in figure legends.
Question 9. Why did the authors present A172 data in Figure 6 followed by U87MG data in Figure 7?! Figure 6 either need to be changed with Figure S4 or they need to show the data for A172 in Figure 7.
Figure 6 was changed with S4 accordingly.
Question 10. The manuscript need grammatical and English language correction as well as correction to the style (bold/un-bold) is used in various sections. There is several errors including the use of comma (,) after full stop (.) and open parenthesis with no closing.
Grammatical and English language was checked by the co-author Dr. David Sherris, native English speaker and researcher. Style and typing errors were corrected.
Question 11. There is also a general inconsistency with the way figures are referred to.
This was fixed.
Question 12. Figure 3, 4, 6, 8 and 9 there is no need for the tables these could be explained in the legend and illustrated on the graphs.
I have eliminated tables showing statistical analyses from the figures. Since description of this statistical data could be confusing, we have generated new figures to be moved in the Supplemental data as follows:
tables old figure 3 as new figure S2
old figure 4 as new figure S3
old figure 6 as new figure S4
old figure 8 as new figure S5
old figure 9 as new figure S6
Question 12. Line 220 the injected number of cells need to be fixed I believe the authors meant 10^3 not 103!
This is a typing error and is corrected.
Question 13. Clarify Line 280, antibodies on 11 our 13 GBM cell lines.
The analyses were performed on 11 of 12 GBM cell lines present in our original cell cohort (lacking SF268 which was not examined).
Question 14. The resolution of Figure 1, 2 and 7 is very low
Images was uploaded at 600 dpi and graphs at least 300 dpi.
Question 15. Mention RES-529 (RES529) in the same form throughout the document.
This was fixed.
Reviewer 2 Report
I would like to thank the authors for taking into account all my comments. Regards
Author Response
I think this reviewer for the good consideration on the work that we did to answer your questions.
Round 3
Reviewer 1 Report
The Authors have addressed my concerns.
Author Response
Dear reviewer
Thanks for your assistance!
Claudio